# Exploring the Profile of Cell Populations and Soluble Immunological Mediators in *Bothrops atrox* Envenomations

**DOI:** 10.3390/toxins15030196

**Published:** 2023-03-04

**Authors:** Kerolaine Fonseca Coelho, Juliana Costa Ferreira Neves, Hiochelson Najibe Santos Ibiapina, Fábio Magalhães-Gama, Fabiane Bianca Albuquerque Barbosa, Flavio Souza Silva, Irmgardt Alicia María Wellmann, Jacqueline Almeida Gonçalves Sachett, Andréa Monteiro Tarragô, Luiz Carlos Lima Ferreira, Adriana Malheiro, Wuelton Marcelo Monteiro, Allyson Guimarães Costa

**Affiliations:** 1Programa de Pós-Graduação em Medicina Tropical, Universidade do Estado do Amazonas (UEA), Manaus 69040-000, AM, Brazil; 2Instituto de Pesquisa Clínica Carlos Borborema, Fundação de Medicina Tropical Doutor Heitor Vieira Dourado (FMT-HVD), Manaus 69040-000, AM, Brazil; 3Programa de Pós-Graduação em Ciências da Saúde, Instituto René Rachou-Fundação Oswaldo Cruz (FIOCRUZ Minas), Belo Horizonte 30190-002, MG, Brazil; 4Diretoria de Ensino e Pesquisa, Fundação Hospitalar de Hematologia e Hemoterapia do Amazonas (HEMOAM), Manaus 69050-001, AM, Brazil; 5Programa de Pós-Graduação em Imunologia Básica e Aplicada, Universidade Federal do Amazonas (UFAM), Manaus 69067-005, AM, Brazil; 6Departamento de Ensino e Pesquisa, Fundação Alfredo da Matta (FUAM), Manaus 69065-130, AM, Brazil; 7Programa de Pós-Graduação em Ciências Aplicadas à Hematologia, Universidade do Estado do Amazonas (UEA), Manaus 69050-001, AM, Brazil; 8Escola de Enfermagem de Manaus, Universidade Federal do Amazonas (UFAM), Manaus 69057-070, AM, Brazil

**Keywords:** *Bothrops* snakebite, immune response, cell populations, chemokines, cytokines, blister

## Abstract

*Bothrops atrox* envenomations are common in the Brazilian Amazon. The venom of *B. atrox* is highly inflammatory, which results in severe local complications, including the formation of blisters. Moreover, there is little information on the immune mechanisms associated with this condition. Thus, a longitudinal study was carried out to characterize the profile of the cell populations and soluble immunological mediators in the peripheral blood and blisters in *B. atrox* patients s according to their clinical manifestations (mild and severe). A similar response in both *B. atrox* patient groups (MILD and SEV) was observed, with an increase in inflammatory monocytes, NKT, and T and B cells, as well as CCL2, CCL5, CXCL9, CXCL10, IL-1β and IL-10, when compared with the group of healthy blood donors. After the administration of antivenom, the participation of patrolling monocytes and IL-10 in the MILD group was observed. In the SEV group, the participation of B cells was observed, with high levels of CCL2 and IL-6. In the blister exudate, a hyperinflammatory profile was observed. In conclusion, we revealed the involvement of cell populations and soluble mediators in the immune response to *B. atrox* envenomation at the local and peripheral level, which is related to the onset and extent of the inflammation/clinical manifestation.

## 1. Introduction

Snakebite envenomations represent a major public health problem and present high morbidity and mortality in tropical and subtropical countries [1,2,3,4]. In Brazil, most of these envenomations are caused by the genus *Bothrops*, with the species *Bothrops atrox* being responsible for 90% of reported cases in the Brazilian Amazon [5,6,7]. In 2021, 9902 snakebite envenomations were reported in this region, of which 8436 were attributed to this species [8]. The venom of *B. atrox* (BaV) consists of enzymes such as metalloproteases (MPs), serinoproteases (SPs) and phospholipases A2 (PLA2s), which promote local and systemic disorders as a result of their coagulotoxic, proteolytic and acute inflammatory activities [4]. The exacerbation of the inflammatory response, in conjunction with the tissue damage caused by BaV, results in the development of severe local complications [9,10].

MPs damage the basement membrane of capillaries and the epidermis, causing separation of the dermal–epidermal junction by degrading Type IV collagen, laminin and other components of the extracellular matrix and plasma membrane [11,12]. This degradation culminates in the structural destabilization of the tissue, causing the weakening and distention of microvessels [13,14,15]. In this context, PLA2s contribute to tissue damage through their myotoxic activity in muscle fibers [16,17]. Together, these events lead to extravasation and the formation of blisters, which is an aggravating factor that facilitates the entry of microorganisms and causes secondary infections, with a frequency of 40% [18,19]. In addition, studies carried out in the Brazilian Amazon observed that 6.6% of *B. atrox* patients evolve blister formation, of which 8.3% present with secondary infections [20]. 

There is also a release of biologically active peptides that induce the inflammatory process, activating endogenous signaling pathways, recruiting leukocytes and releasing numerous mediators [19,21,22]. 

For the maintenance of homeostasis after tissue injury, the coordinated activation of the inflammatory process is essential [23,24]. Several soluble immune mediators are associated with the inflammation promoted by snake venom, such as the cytokines TNF-α, IL-6, IL-12p70 and IL-10, and the chemokine CCL2 [25,26,27]. A study carried out in the Brazilian Amazon on patients bitten by *B. atrox* showed that, before antivenom therapy, there was an increase in the serum levels of CXCL-9, CXCL10, IL-6 and IL-10 [28]. Regarding the involvement of cell populations, studies have described an initial increase in polymorphonuclear cells, followed by an increase in mononuclear cells, with the action of monocytes/macrophages, dendritic cells (DC) and natural killer cells (NK), in addition to the participation of TCD4+ cells, TCD8+ cells and B cells, thus playing an important role in pathogenesis, with a complex and poorly explored response repertoire in humans [26,29,30]. Thus, this study sought to explore and describe the cellular profile and immunological mediators involved in *B. atrox* envenomation, which may help us to understand the pathogenesis and response to antivenom treatment in this clinical condition.

## 2. Results

### 2.1. Characterization of the Study Population and Laboratory Data

The main clinical and epidemiological characteristics of the study population are described in Table 1. In the MILD and SEV groups, there was a predominance of males, and the groups had a median age of 46 and 53, respectively, for MILD and SEV. The median age of the healthy blood donors (HD) group was 33 years. Most patients had had no previous snakebite envenomations, and most were mainly residents of rural areas. The lower limbs were the most affected, and the envenomations were mostly associated with work activities. Adverse reactions to the antivenom were greater in the SEV group, which was the same group that presented with a longer time between the moment of the bite and hospital admission. There was a significant difference in age, the affected anatomical site and the frequency of adverse reactions to the antivenom between the MILD and SEV groups (*p* = 0.0067, *p* = 0.010, *p* = 0.035, respectively).

### 2.2. Blood Count and Immunophenotyping Data in Bothrops atrox Snakebite Patients, Stratified into Mild and Severe Groups, before and after the Administration of Antivenom 

To evaluate and compare the blood count and immunophenotyping values in *B. atrox* snakebite patients, which were stratified into the MILD and SEV groups, an intragroup and intergroup analysis of the hematological data was performed before and after the administration of antivenom, as described in Table 2. In general, the intragroup analysis showed a significant decrease in the values of the red and white series, and platelets at 48 h after serum therapy (T2) compared with the values before serum therapy (T0) and 24 h after serum therapy (T1) in both the MILD and SEV groups. 

### 2.3. Snakebite Patients Are Marked by an Inflammatory Profile That Is Driven by a Substantial Increase in Immunological Mediators

The immunophenotyping profile of *B. atrox* snakebite patients is presented in Figure 1A. Lower frequencies of classical monocytes, NK cells, plasmacytoid DC and helper T cells were observed in *B. atrox* snakebite patients (MILD and SEV) compared with the healthy donors (HD). On the other hand, inflammatory monocytes, NKT cells, and B1 and T cells were increased in *B. atrox* snakebite patients. In addition, the data demonstrated a decrease in classical monocyte and B cell populations in the MILD group compared with the SEV group. The profile of serum chemokines and cytokines observed in *Bothrops* snakebite patients are shown in Figure 1B. A general increase in the levels of the evaluated soluble immunological mediators, including CCL2, CXCL9, CCL5, CXCL10, IL-1β, IL-6 and IL-10, along with a unique decrease in TNF, was observed in *B. atrox* snakebite patients compared with the HD group.

### 2.4. Follow-Up of the Clinical Course of Mild and Severe Bothrops atrox Snakebite Patients Demonstrated Distinct Immunological Profiles at T2

To assess the dynamics of the immune response after antivenom therapy, immunophenotypic characterization was performed and the levels of immunological mediators were evaluated at T1 (24 h after antivenom therapy) and T2 (48 h after antivenom therapy), in the MILD and SEV patients, and are presented in Figure 2A,B, respectively. At T1, there were few differences between the groups, which are represented by the increase in patrolling monocytes and a decrease in IL-6 in the MILD group. In contrast, a change in this behavior was observed at T2, since, compared with the SEV group, the MILD group showed a significant increase in the frequency of classical and inflammatory monocytes, NKT cells, and cytotoxic T and T cells, as well as in the levels of CXCL-9, CXCL-10, IL-1β, TNF and IL-10. In addition, a reduction in the frequency of patrolling monocytes and B cells was observed, along with the mediators CCL2 and IL-6.

### 2.5. Snakebite Patients Exhibit Complex Integrative Networks and Interactions between the Elements of Peripheral Blood and Blister Exudate

Integrative networks were constructed to demonstrate the interaction between the cell phenotypes and the immunological mediators at T0, T1 and T2, as shown in Figure 3. HDs were used as a reference group and demonstrated a restricted network of interactions between the immunological mediators, with a small number of neighborhood connections, together with cell populations marked by negative correlations. On the other hand, the MILD group presented a network rich in interactions at T0, with a high number of connections between chemokines and pro-inflammatory cytokines, together with the regulatory cytokine IL-10. The SEV group exhibited a distinct profile, with shading of the interactions between immunological mediators, and the presence of negative correlations.

### 2.6. Evaluation of Blister Exudate in Severe Bothrops atrox Snakebite Patients Revealed Cellular Infiltration and Inflammatory Status

To obtain a broader view of the immune response, blister exudate from patients with severe tissue complications were evaluated at T2 and compared with peripheral blood, as shown in Figure 4. The data demonstrated infiltration of cell populations, with a significant increase in the frequency of inflammatory and patrolling monocytes, NKT cells and helper T cells (Figure 4A). In addition, the blister exudate also demonstrated higher levels of inflammatory mediators, with a significant increase in CCL2, IL-1β, IL-6, TNF and IL-10 (Figure 4B). On the other hand, classical monocytes, T cells and the chemokine receptor CCL5 were higher in the peripheral blood.

### 2.7. Snakebite Patients Exhibit Complex Integrative Networks and Interactions among the Elements of Peripheral Blood and Blister Exudate

After antivenom therapy, at T1 and T2, it was possible to observe that the MILD group maintained a network profile similar to that of T0, with a slight decrease in the connections among pro-inflammatory mediators. Likewise, at T2, the SEV group exhibited a profile similar to that observed at T0, which was characterized by the presence of negative correlations among the cell populations and immunological mediators. Networks were also built for patients with severe tissue complications, with the aim of investigating the interactions among the elements of the blister exudate, and the communication with the elements of the peripheral blood, as presented in Figure 5. The results showed that, in general, the blister compartment was characterized by the absence of connections among the elements. However, it was possible to observe a clear interaction among the cell populations and the immunological mediators of the blister exudate and peripheral blood, thus indicating a process of recruitment to the inflammatory site.

## 3. Discussion

Tissue complications caused by *Bothrops* envenomation directly influence the prognosis of patients, and clinical and epidemiological factors are relevant in terms of aggravation of the outcome, with advanced age, and the absence of or delays in treatment being associated with the severity [33,34]. This corroborates our findings, as the SEV group showed a greater mean age and a longer time until arrival at the medical center. The high number of adverse reactions found in the SEV group may be related to the number of vials administered during treatment after classifying the accident [32,35] (Table 1). Changes in the hematological values were also observed in patients, with a hemorrhagic process leading to a decrease in erythrocytes, hemoglobin and platelets (Table 2). Clinically, the reductions in these parameters were correlated with the severity of the envenomation, and thrombocytopenia on admission was linked to the intense local and systemic hemorrhagic condition that is characteristic of victims of severe snakebite [36,37,38]. 

Neutrophilic leukocytosis with a leftward shift is also associated with the extent of tissue damage in patients, [25,39,40], and this was corroborated by the data observed in the hemogram, which showed an increase in the neutrophil count. Neutrophils are the first inflammatory cells to reach the damaged tissue, in addition to acting directly on infectious complications after the bite, and induce the synthesis of inflammatory mediators [41,42,43,44,45]. A significant decrease in neutrophils at T2 was also observed in both study groups (Table 2). They are also involved in adaptive immunity by regulating T cells’ functions [43]. Experimental studies have shown a rapid inversion in the ratio of neutrophils to lymphocytes, which shows that *Bothrops* venom promotes early immune changes, and the proportion of lymphocytes has been used as a predictor of severity in injured patients [11,46,47,48]. Similarly, it was observed that *B. atrox* snakebite patients presented lymphopenia compared with the control group, markedly in the SEV group, thus reinforcing its correlation with severity.

Some studies have reported that the mediators produced by neutrophils pave the way for the recruitment of inflammatory monocytes [49,50,51]. Monocytes are mononuclear phagocytes that, when recruited, differentiate into macrophages and secrete numerous pro-inflammatory cytokines, thereby intensifying leukocyte recruitment. Our results demonstrated an increase in the populations of inflammatory monocytes, NKT cells and B1 and T lymphocytes, along with increased production of chemokines and inflammatory cytokines (Figure 1A,B).

NKT cells correspond to populations of innate-like T cells that are rapidly activated and can potentiate the action of DCs, macrophages and B lymphocytes, or suppress the immune responses [52]. During patient follow-up, an increase in the frequency of NKT cells was observed in the MILD group (T1) (Figure 2A). A profile change was observed at T2, and compared with the SEV group, the MILD group showed a significant increase in the levels of CXCL9, CXCL10, IL-1β, TNF and IL-10 (Figure 2B). 

The cytokine IL-10 is characterized as a key molecule in the regulation of the immune response, which is capable of suppressing the activation of neutrophils and monocytes, and, consequently, the production of chemokines and cytokines, thus regulating the inflammatory processes and cytotoxic pathways [53,54]. Our data from the integrative analysis of the immunological molecules demonstrated the effective participation of IL-10, with several interactions established before and after the administration of antivenom in the MILD group compared with the SEV group, which may be related to the lower severity in these patients (Figure 3). In additional, by comparing the cells and mediators present in the exudate and peripheral blood, we noticed an increase in the recruitment not only of NKT cells but also of inflammatory and patrolling monocytes, in addition to the CCL2, IL1-β, IL-6, TNF and IL-10 mediators, reflecting the local hyperinflammatory process (Figure 4A,B). This distinct response profile between the groups highlights the importance of IL-10 in controlling the immune response to venom toxins. Furthermore, we observed correlations among the molecules and cells present in the peripheral blood and in the blisters, which indicated the interactions among the components of the peripheral blood and the blister exudate (Figure 5).

Based on our findings, a schematic model of the local and systemic inflammatory response was constructed (Figure 6). Before the administration of antivenom, we could see a similar response pattern between the groups, with an increase in the percentage of inflammatory monocytes, T lymphocytes, NKT and B cells, together with the chemokines CCL2, CXCL9, CCL5 and CXCL10, and the cytokines IL-1β and IL-10. After the administration of antivenom, we observed the participation of the patrolling monocytes and IL-10 in the MILD group. On the other hand, in the SEV group, the participation of the chemokine CCL2 was observed, which showed high concentrations, both in the circulation and in the exudate. CCL2 is one of the main chemoattractants of monocytes and has previously been described as a potential biomarker of severe tissue complications and systemic complications such as acute renal failure [55]. Finally, the blister exudate was characterized by hyperinflation, marked by the migration of classical and inflammatory monocytes with NKT and Th cells, together with elevated levels of IL-1β, IL-6, TNF and IL-10.

It is important to note that the study has certain limitations such as the short period of recruitment and the low frequency of blister development, reflecting the decrease in the number of samples. We emphasize that the short period of hospitalization (3 days on average, which reflects the efficiency of the health system) made a more extensive follow-up unfeasible. In addition, it was not possible to assess the presence of neutrophils in the peripheral blood and blister exudate via immunophenotyping. However, the results presented here may help us to understand the local and systemic immune response of these envenomations, which are of great clinical and epidemiological relevance. It is also necessary to highlight the difference in age between the control group (lower mean age) and the injured group (higher mean age), which may have an impact on the state of the immune system of these participants. Therefore, there is an evident need for additional studies that further evaluate the immune response of *B. atrox* snakebite victims, since most of the studies described in the literature have been performed in vitro or in an animal model.

## 4. Conclusions

The exacerbation of the inflammatory process after *B. atrox* snakebites has been studied in in vitro and in vivo experiments, and can be used to define the prognosis of the envenomations. Cells such as neutrophils and monocytes play a central role in the body’s defense; however, this same defensive role, if not properly regulated, can lead to the development of local and systemic complications. Our results explored and demonstrated that cell populations and soluble mediators are important components of this inflammatory response, with a distinct profile seen between the groups after the snakebites. The information present in this study clarifies the mechanism of the inflammatory response after *Bothops* snakebites. We suggest that further studies be carried out to identify the molecules or cells that may be predictive biomarkers in these patients.

## 5. Materials and Methods

### 5.1. Study Population and Design

The study population consisted of 48 individuals diagnosed with a *B. atrox* envenomation after seeking medical care at the Fundação de Medicina Tropical Dr. Heitor Vieira Dourado (FMT-HVD) between March 2019 and March 2021. The identification of *B. atrox* was performed by a zoologist from the research group.

The patients included in the study were classified according to the Brazilian Ministry of Health’s guidelines [55] into two groups, referred to as mild (MILD) and severe (SEV). The MILD group was composed of patients who did not develop severe local and systemic manifestations in the first 48 h after hospitalization, and were treated with 2 to 8 vials of Bothrops antivenom. The severe group (SEV) included patients with severe local and systemic changes, with severe tissue complications and blistering, and were treated with 12 vials of Bothrops antivenom, as shown in Figure 7. The patients were aged over 18 years, had not received previous serum therapy in another health unit, and agreed to participate in the study by signing the informed consent form.

Pregnant women, individuals under the age of 18 or those who reported having a history of previous inflammatory diseases, such as diabetes, autoimmune diseases or immunodeficiency, individuals using platelet antiaggregants and/or anticoagulants, and those who arrived at the emergency clinic 24 h after the snakebite were not included. In addition, in partnership with the Fundação Hospitalar de Hematologia e Hemoterapia do Amazonas (HEMOAM), 48 healthy blood donors (HD) of both sexes with no history of snakebite were included as the control group. 

### 5.2. Ethical Issues

This study was submitted to and approved by the Ethical Committee at FMT-HVD, with the protocol registration number #492,892. All the participants read and signed an informed consent form. The procedures performed are in accordance with the Declaration of Helsinki and Resolution 466/2012 of the Brazilian National Health Council for research involving human participants.

### 5.3. Collection of Biological Samples and Acquisition of Clinical and Laboratory Data

Patients were monitored and had their peripheral blood collected at 3 time points, referred to as T0 (before antivenom therapy), T1 (24 h after antivenom therapy) and T2 (48 h after antivenom therapy). In addition, blister exudate from patients with severe tissue injury was collected at T2. The peripheral blood and blister exudate were obtained via venipuncture and aspiration, respectively, with 4 mL of peripheral blood (BD Vacutainer EDTA K2) and 200–300 μL of exudate collected from each patient. Full blood counts were performed at T0, T1 and T2 in the hematology sector of the Fundação HEMOAM using the ADVIA 2120i Hematology System (Siemens Healthcare Diagnosis). The collected samples were stored at room temperature (20 °C to 25 °C) and processed within 4 h of collection. Examinations of the complementary and inflammation markers were not requested as part of the medical routine during the patients’ hospitalization period; therefore, our study did not collect these data.

There was no addition of anticoagulants. After blood collection and counting, samples were submitted to centrifugation at 900× *g*, for 15 min. The supernatant was collected and immediately stored at −80 °C until the evaluation of the soluble immunological mediators. Blister exudate was also submitted to centrifugation at 900× *g* for 15 min, and the supernatant was also collected and stored at −80 °C until the evaluation of the soluble immunological mediators. Subsequently, the volume of the supernatant of blood samples and blister exudate was made up with PBS and homogenized for cell immunophenotyping. Patients answered a questionnaire with sociodemographic and epidemiological variables to obtain their clinical and epidemiological data, and the clinical information present in the electronic medical records (iDoctor) of the FMT-HVD was also used.

### 5.4. Immunophenotypic Characterization and Analysis

The immunophenotypic characterization was performed using flow cytometry. The cells were obtained from an aliquot of 100 μL from the peripheral blood and were incubated at room temperature (20 °C to 25 °C) for 30 min in the presence of the following fluorescent-labeled specific anti-human cell surface monoclonal antibodies: anti-CD14-APC/CD16-FITC/HLA-DR-PE to identify the monocyte subtypes (classical, inflammatory and patrolling), anti-CD3-PECy7/CD16-FITC/CD56-PE to identify the natural killer cells (NK) and natural killer T cells (NKT), anti-CD14-APC/CD11c-PE/CD123-FITC to identify the classical and plasmacytoid dendritic cells (DC), anti-CD19-FITC/CD5-PE to identify the B and B1 cells, and anti-CD3-FITC/CD4-PECy7/CD8-PE to identify the helper T cells and cytotoxic T cells. Antibodies were purchased from BD Biosciences (San Diego, CA, USA), Beckman Coulter (Brea, CA, USA) and BioLegend (San Diego, CA, USA). It is worth noting that for labeling the cells of the blister exudate, the same labeling panel was used but with the implementation of the pan-leukocyte antibody (anti-CD45-PerCP) to verify whether the labeled cells corresponded to the leukocyte population of interest. After incubation, the cells were treated with 2 mL of an erythrocyte lysing solution (BD FACS Lysing Solution, BD Biosciences San Diego, CA, USA) for 10 min at room temperature. After two centrifugation steps and two washing steps with PBS, the cells were resuspended in 300 μL of PBS for acquisition in the cytometer. 

Sample acquisition was performed on the FACSCanto II cytometer (BD Biosciences, San Jose, CA, USA) at HEMOAM. In total, 30,000 events were acquired for each sample to quantify the cell populations. Furthermore, for the morphometric and immunophenotypic identification of the cells, FlowJo Software (v10.8) was used, with the design of the gates set to select the target populations in graphs that combined the morphological characteristics (size and granulosity) with the immunophenotypic characteristics through the fluorescence of the monoclonal antibodies used to identify the target cells.

### 5.5. Quantification of the Soluble Immunological Mediators

The levels of the soluble immunological mediators, including chemokines (CXCL8, CCL2, CXCL9, CCL5 and CXCL10) and cytokines (IL-1β, IL-6, TNF and IL-10), were quantified using cytometric bead arrays (BD Human Chemokine and BD Human Inflammatory Cytokines kits, BD Biosciences, San Diego, CA, USA), according to the manufacturer’s instructions. Samples were acquired in the FACSCanto II cytometer (BD Biosciences, San Jose, CA, USA) at HEMOAM. FCAP-Array software v3 (BD Biosciences, San Jose, CA, USA) was used to calculate the chemokine and cytokine levels. Data were reported in picograms per milliliter (pg/mL) concentrations, according to the standard curves provided in the kits.

### 5.6. Conventional Statistical Analysis

The Shapiro–Wilk test was performed for each variable to verify the data’s normality and revealed a non-parametric distribution. The comparisons of the values between two independent groups were performed using the Mann–Whitney test, while comparisons between two dependent groups were performed using Wilcoxon’s matched pairs signed rank test. In addition, multiple comparisons among groups were performed using Kruskal–Wallis tests followed by Dunn’s test. In all cases, significance was considered at *p* < 0.05. GraphPad Prism v8.0.1 (GraphPad Software, San Diego, CA, USA) software was used for the statistical analyses. GraphPad Prism and PowerPoint software were used for graphical figures.

### 5.7. Correlation Network Analysis

The multiple correlations among the cell populations and the soluble immunological mediators in the patients of *Bothrops* snakebites were determined by using Spearman’s correlation coefficient in GraphPad Prism v8.0.1 (GraphPad Software, San Diego, CA, USA). Positive and negative correlations were considered significant when *p* < 0.05. Following this, the significant correlations were compiled using the open source Cytoscape software (v3.0.3, National Institute of General Medical Sciences, Bethesda, MD, USA). Immunological networks were constructed using circular layouts, in which each cell or mediator was represented by a globular node. The correlation index (r) was used to categorize the strength of a correlation as weak (r ≤ 0.35), moderate (r ≥ 0.36 to r ≤ 0.67) or strong (r ≥ 0.68), which was represented by connecting edges [31].

## Figures and Tables

**Figure 1 toxins-15-00196-f001:**
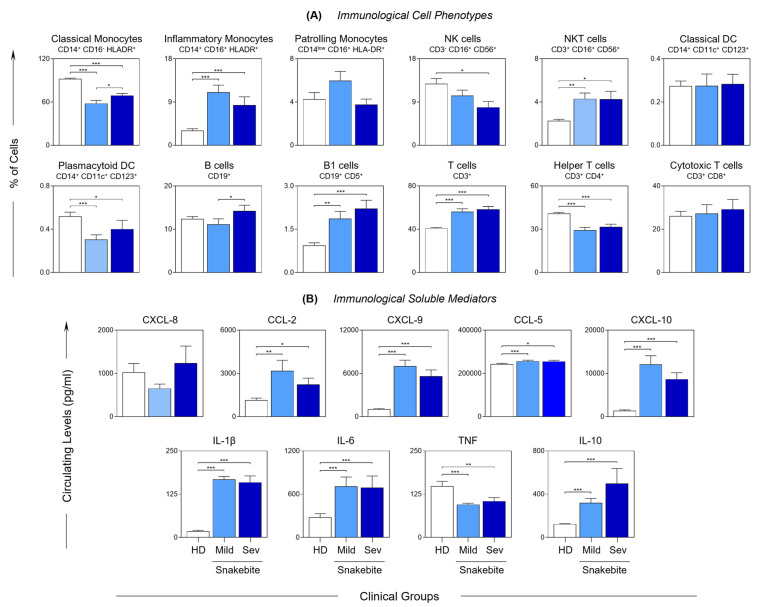
**Immunological profile in *Bothrops* snakebite patients before the administration of antivenom.** Cell phenotypes (**A**) and immunological soluble mediators (**B**) were evaluated before antivenom administration (T0) in the peripheral blood of *Bothrops* snakebite patients, stratified into mild (
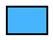
) and severe (
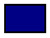
) groups, and in healthy donors (
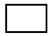
). The frequency of the cell populations and the levels of the soluble immunological mediators were measured using flow cytometry and CBA, respectively. The results are presented using bar plots, with a linear scale. Statistical analyses were performed using the Mann–Whitney test. Significant differences are indicated by connecting lines and asterisks for *p* < 0.001 (***), *p* < 0.01 (**) or *p* < 0.05 (*).

**Figure 2 toxins-15-00196-f002:**
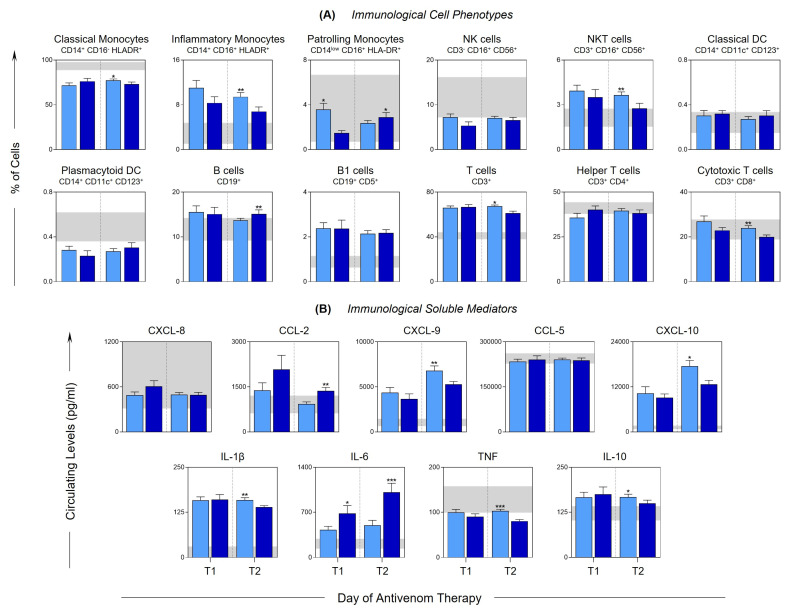
**Dynamics of the immunological profile of *Bothrops* snakebite patients after the administration of antivenom.** Cell phenotypes (**A**) and immunological soluble mediators (**B**) were evaluated at T1 (24 h after antivenom) and T2 (48 h after antivenom) in the peripheral blood of *Bothrops* snakebite patients stratified into mild (
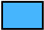
) and severe (

) groups. The frequency of cell populations and the levels of soluble immunological mediators were measured using flow cytometry and CBA, respectively. The results are presented using bar plots, with a linear scale. The interquartile ranges (25–75°) for the results observed in the control group were used as a reference interval (gray background—
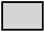
). Statistical analyses were performed using the Mann–Whitney test. Significant differences are indicated by asterisks for *p* < 0.001 (***), *p* < 0.01 (**) and *p* < 0.05 (*).

**Figure 3 toxins-15-00196-f003:**
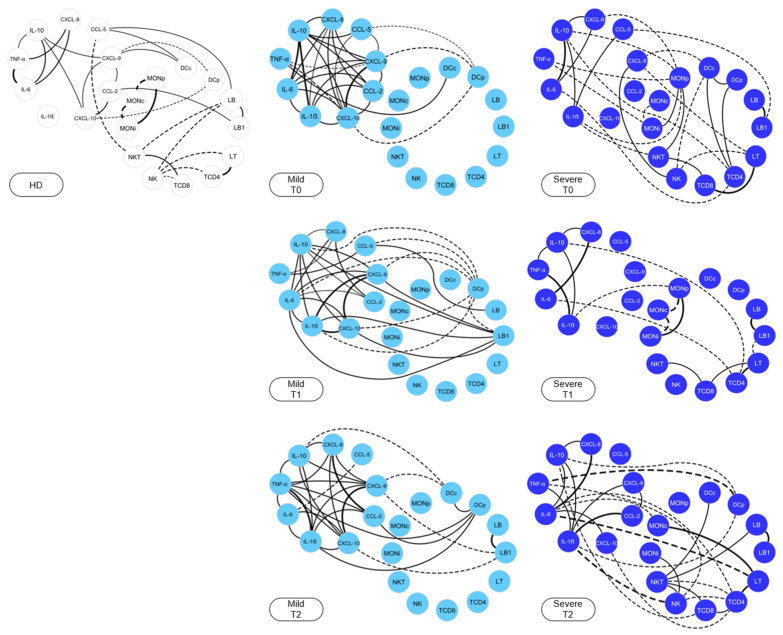
**Integrative networks of *Bothrops* snakebite patients in the peripheral blood samples.** Networks of *Bothrops* snakebite patients were constructed before (T0) and after (T1 and T2) the administration of antivenom, demonstrating the interactions between cell phenotypes and immunological soluble mediators. White nodes (
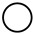
) are used to identify healthy donors, light blue nodes (
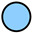
) are used to identify the MILD group, while dark blue (
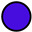
) is used to identify the SEVERE group. Solid lines between the elements indicate a positive correlation, while dashed lines indicate a negative correlation. The thickness of the lines indicates the strength of the correlation. The correlation index (r) was used to categorize the strength of the correlation as weak (r ≤ 0.35), moderate (r ≥ 0.36 to r ≤ 0.67) or strong (r ≥ 0.68).

**Figure 4 toxins-15-00196-f004:**
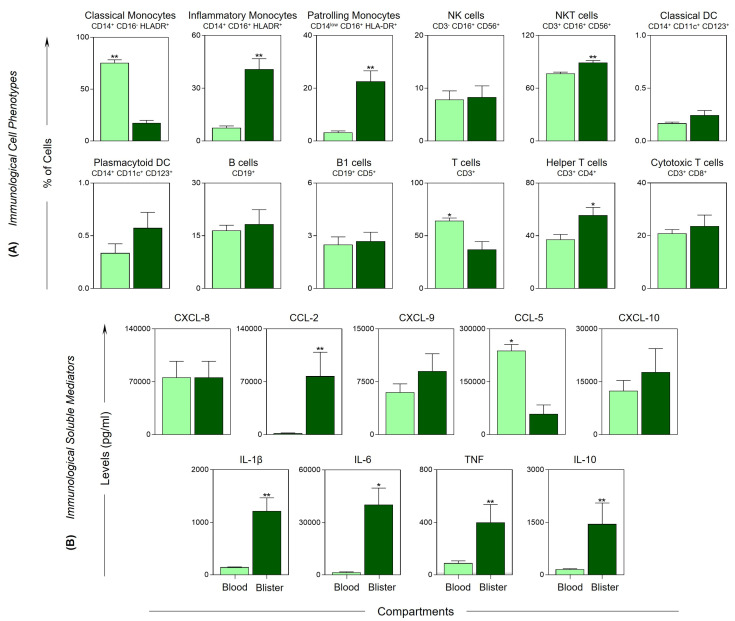
**Comparison of blister exudate vs. peripheral blood in severe *Bothrops* snakebite patients.** Cell phenotypes (**A**) and immunological soluble mediators (**B**) were evaluated at T2 (48 h after the administration of antivenom) in the blood (
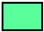
) and blister (

) compartment of severe *Bothrops* snakebite patients. The frequency of cell populations and the levels of soluble immunological mediators were measured using flow cytometry and CBA, respectively. The results are presented using bar plots, with a linear scale. Statistical analyses were performed using Wilcoxon’s matched pairs signed rank test. Significant differences are indicated by asterisks for *p* < 0.01 (**) and *p* < 0.05 (*).

**Figure 5 toxins-15-00196-f005:**
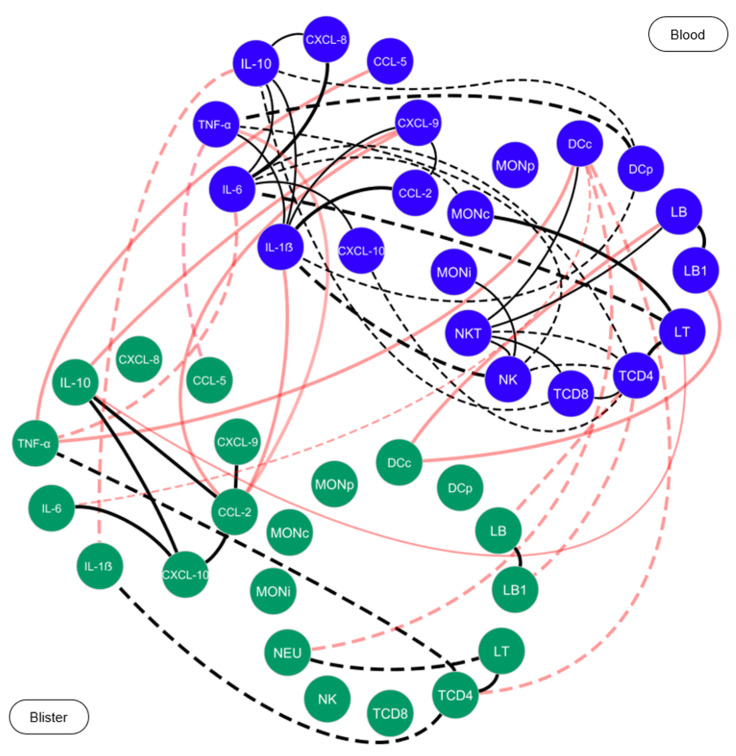
**Integrative networks of *Bothrops* snakebite patients in the peripheral blood and blister exudate.** Networks of *Bothrops* snakebite patients were constructed before (T0) and after (T1 and T2) the administration of antivenom, demonstrating the interactions of cell phenotypes and immunological soluble mediators in blisters and peripheral blood. Green nodes (
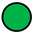
) represent the elements of blister exudate, and blue nodes (
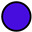
) are used to identify the elements of peripheral blood. Solid lines between the elements indicate a positive correlation, while dashed lines indicate a negative correlation. The red lines indicate the correlations between the blister exudate and peripheral blood. The thickness of lines indicates the strength of the correlation. The correlation index (r) was used to categorize the strength of the correlation as weak (r ≤ 0.35), moderate (r ≥ 0.36 to r ≤ 0.67) or strong (r ≥ 0.68).

**Figure 6 toxins-15-00196-f006:**
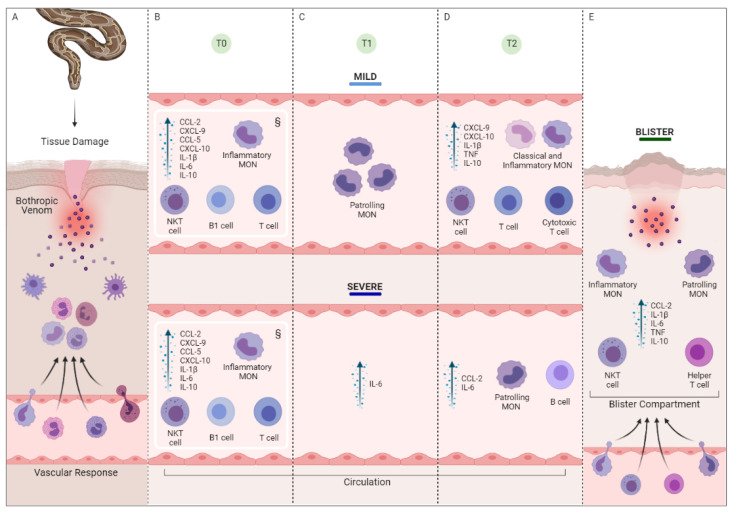
**Schematic summary of the local and systemic inflammatory response to *B. atrox* envenomation with and without severe tissue complications.** Quadrant (**A**) represents the bite and subcutaneous/intramuscular injection of BaV, inducing tissue damage and promoting the activation of the resident cells, the vascular response and the recruitment of circulating cells to the affected site. (**B**) Pre-serum therapy response of patients stratified into mild and severe groups, compared with the control group, represented by the symbol (§). The case groups show a similar pattern of response at T0, with an increase in the frequency of inflammatory monocytes, T lymphocytes, NKT and B cells. (**C**) Changes in response profile 24 h after the administration of serum therapy. The possible action of patrolling monocytes was seen in the mild group and the inflammatory cytokine IL-6 in the severe group. (**D**) The percentage of cells increased 48 h post-treatment in the MILD group for classic and inflammatory monocytes and TCD8+ lymphocytes, in addition to CXCL9, CXCL10, IL1β, TNF and IL-10. In the SEV group, the response profile demonstrated the possible action of patrolling monocytes, B lymphocytes and CCL2 and IL-6 molecules. Quadrant (**E**) represents the response at the local level, with the action of subpopulations of the inflammatory and patrolling monocytes, as well as NKT and TCD4+ lymphocytes, and the CCL2, IL1 β, IL-6 and IL-10 mediators.

**Figure 7 toxins-15-00196-f007:**
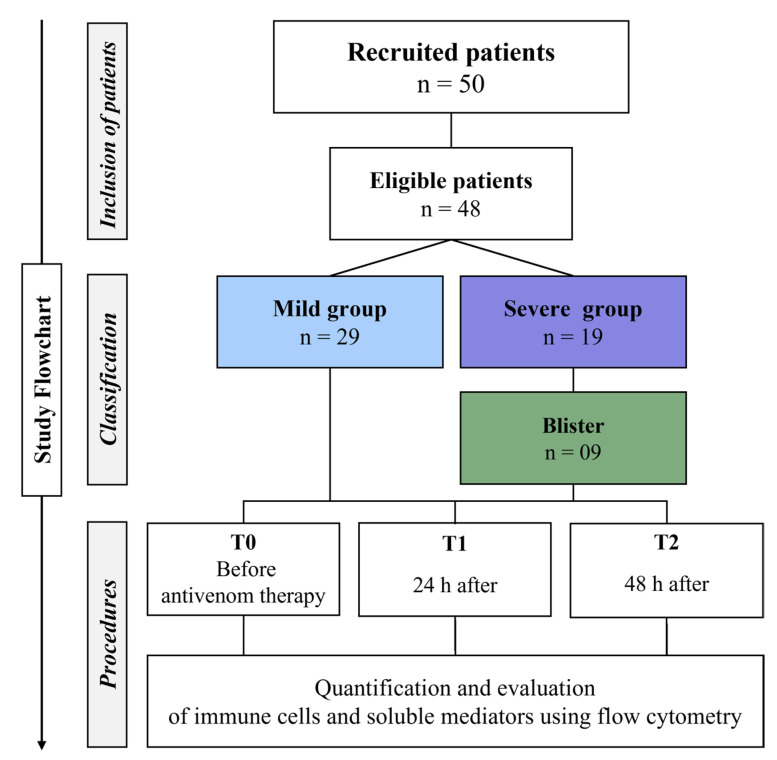
**Study flowchart.** In total, 48 patients were considered eligible, which were divided into two groups according to their clinical evolution and were classified as either mild or severe. The mild group was composed of 29 patients and the severe group comprised 19 patients. In addition, of the patients with a severe snakebite, 9 had tissue complications and blister formation, and the transudate was collected 48 h after administration of the antivenom. Subsequently, the cellular profile and soluble immunological mediators were evaluated via immunophenotyping and CBA, respectively, before the administration of antivenom and 24 and 48 h after administration.

**Table 1 toxins-15-00196-t001:** Clinical and demographic data of *Bothrops* snakebite patients, stratified into mild and severe groups.

Demographic and Clinical Characteristics	HD (n = 48)	*B. atrox* Patients		*p*-Value
		Mild (n = 29)	Severe (n = 19)	
**Gender (n, male/female)**	34/14	26/3	15/4	0.248
**Age (median and (IQR))**	33(23–44)	46 (28–60)	53 (30–63)	**0.0067** ^#^
**Previous snakebite (n, yes/no)**	-	3/26	5/14	0.146
**Zone of occurrence (n, rural/urban)**	-	26/3	15/4	0.304
**Anatomical site of the snakebite (n, upper/lower) limb)**	-	4/25	9/10	**0.010** ^#^
**Anatomical site of the snakebite (n, right/left)**	-	11/18	12/7	0.087
**Work-related accident (n, yes/no)**	-	10/19	7/12	0.867
**Adverse reaction to the antivenom (n, yes/no) ***	-	8/21	11/8	**0.035** ^#^
**Time between snakebite and administration of the antivenom (hours, median (IQR))**	-	3 (2–6)	6 (3–9)	0.129

HD, healthy blood donor; IQR, interquartile range. Significant differences at *p* < 0.05 for comparisons between the mild and severe groups are represented in bold with the superscript symbol ^#^. * Adverse reactions to antivenom, found in the mild and severe groups, were classified as anaphylactic clinical reactions, and reactions described in the literature as mild or rare events, such as pruritus and skin reactions, hyperemia and nausea [31,32].

**Table 2 toxins-15-00196-t002:** Blood count and immunophenotyping data of *Bothrops* snakebite patients, stratified into mild and severe groups, before and after serum therapy.

Parameters	Mild (n = 29)			Severe (n = 19)			HD (n = 48)
	T0	T1	T2	T0	T1	T2	T0
**Red blood cells**	5.2 (4.8–5.3)	4.6 (4.4–4.9) ^a^	3.2 (2.5–4.5) ^a.b^	5.2 (4.9–5.4)	4.8 (4.6–5.1) ^a^	3.7 (0.7–4.7) ^a.b^	5.0 (4.6–5.4)
**Hemoglobin**	15.0 (14.3–15.4)	13.6 (12.1–14.2) ^a^	8.8 (6.8–13.0) ^a.b^	15.2 (14.5–15.6)	13.9 (13.2–15.4) ^a^	11.4 (1.9–13.5) ^a.b^	14.9 (13.6–16.0)
**Platelets**	221 (167–259) ^§^	191 (182–232)	117 (97–171) ^a.b.#^	185 (149–231) ^§^	202 (151–243)	61 (31–193) ^a.b^	243 (211–282)
**Total leukocytes**	10.9 (8.4–14.6) ^§^	11.6 (8.7–15.0)	4.8 (3.8–7.6) ^a.b^	13.3 (9.7–15.6) ^§^	12.3 (9.1–15.0)	8.3 (1.3–11.5) ^a.b^	14.9 (13.6–16.0)
**Neutrophils**	77.0 (67.7–88.9) ^§^	67.8 (62.9–77.8) ^a^	47.4 (36.8–62.3) ^a.b^	80.3 (69.0–89.0) ^§^	71.0 (64.4–82.9)	60.2 (10.0–77.7) ^a.b^	56.1 (50.0–61.5)
**Lymphocytes**	13.0 (5.0–21.3) ^§^	16.9 (13.3–25.5) ^a^	13.6 (11.3–20.5) ^#^	12.1 (5.3–20.3) ^§^	15.5 (10.0–22.0) ^a^	8.5 (1.3–10.7) ^b^	29.7 (26.4–34.6)
**Monocytes**	4.0 (3.5–5.7) ^§^	5.6 (4.5–6.7) ^a^	4.1 (3.3–5.7) ^b^	4.1 (3.1–5.1) ^§^	5.4 (4.0–6.6)	4.4 (0.7–5.3) ^b^	6.0 (5.1–6.6)
**Eosinophils**	2.0 (0.9–4.7) ^§^	1.8 (1.0–2.6)	2.8 (2.2–3.2) ^b^	1.8 (0.4–5.8) ^§^	2.3 (1.2–3.5)	1.8 (1.0–4.9)	3.3 (2.3–6.1)
**Basophils**	0.4 (0.2–0.8)	0.3 (0.2–0.4) ^a^	0.3 (0.2–0.3) ^#^	0.3 (0.2–0.4) ^§^	0.3 (0.2–0.4)	0.1 (0.0–0.3) ^a.b^	0.5 (0.3–0.9)

T0, before serum therapy; T1, 24 h after serum therapy; T2, 48 h after serum therapy; HD, healthy donor; IQR, interquartile range. Reference values: red blood cells, 4.5–6.0 × 10^6^/µL); hemoglobin, 12–18 g/dL; platelets, 130–400 × 10³/µL; total leukocytes, 5.2–12.4 × 10³/µL; neutrophils, 1.9–8.0 × 10³/µL; lymphocytes, 0.9–5.2 × 10³/µL; monocytes, 0.16–1 × 10³/µL; eosinophils, 0.1–5.0 × 10³/µL; basophils, 0.0–2.0 × 10³/µL. Significant differences at *p* < 0.05 for intragroup analysis are represented in bold with the superscript letters “^a^” and “^b^” refer to comparisons with T0 and T1, respectively. Significant differences at *p* < 0.05 for intergroup analysis are represented in bold with the superscript symbols ^#^ and ^§^, which refer to comparisons with the severe group and the healthy donor group, respectively.

## Data Availability

The data used to support the findings of this study are included within the article.

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
