# Peer review of "Exploring the Profile of Cell Populations and Soluble Immunological Mediators in Bothrops atrox Envenomations"

_toxins, 2023, doi:10.3390/toxins15030196_

Round 1
Reviewer 1 Report
The paper entitled “Exploring the profile of cell populations and soluble immunological mediators in Bothrops atrox envenomations” includes potentially relevant data for general medicine, family medicine, pharmacognosy, phytotherapy and potentially for the development and implementation of alternative therapeutic algorithms focused on acute poisoning toxicology. The Authors of the manuscript in question tend to suggest that ..”involvement of cell populations and soluble mediators in the immune response to Bothrops atrox envenomation at the local and peripheral level, which is related to the onset and extent of inflammation/clinical manifestation..”.
Remarks:
1. The introduction needs to prepared on the basis on the up-to-date literature items;
2. Conclusions should represent the answer to the research question/tasks specified in the purpose/objectives of the manuscript. However, it is difficult to find a precisely defined purpose of the work in the manuscript of the publication sent to the TOXINS editorial office. The purpose/aim of the study should be precisely defined and supplemented in the manuscript;
3. In the Materials and Methods section the inclusion and study inclusion criteria were not clearly included. The lack of mentioned criteria may call into question the control of the test results obtained.
4. Test results do not include routine markers of inflammation such as ESR or CRP. The results of the study do not include enzymatic markers describing the functional state of the liver. It is absolutely necessary to supplement the reviewed manuscript with the mentioned the result of routine laboratory tests or explain the absence of the test results in question;
5. The manuscript in question omits the practical/clinical implications of the described in this publication results.
Author Response
Response Letter
February 18th, 2023
To: Academic Editor
Toxins
Section: Animal Venoms
Special Issue: Venoms and Toxin-Mediated Local Manifestations
Dear Editor,
We very much appreciate the kind consideration given to our manuscript entitled “Exploring the profile of cell populations and soluble immunological mediators in Bothrops atrox envenomation. We hope that the replies to the reviewer’s comments will have satisfactorily improved the manuscript.
Below, we present all the queries made by the reviewers. The changes requested are clearly outlined in the revised manuscript and marked in green. We have prepared a list of answers to the reviewers’ comments, which are highlighted in “bold italic”. In the response to each query, we also included the location of the modified part in the revised manuscript.
Reviewers' comments:
Reviewer reports:
Reviewer #1:
The paper entitled “Exploring the profile of cell populations and soluble immunological mediators in Bothrops atrox envenomations” includes potentially relevant data for general medicine, family medicine, pharmacognosy, phytotherapy and potentially for the development and implementation of alternative therapeutic algorithms focused on acute poisoning toxicology. The Authors of the manuscript in question tend to suggest that ..”involvement of cell populations and soluble mediators in the immune response to Bothrops atrox envenomation at the local and peripheral level, which is related to the onset and extent of inflammation/clinical manifestation..”.
We acknowledge the Reviewer #1 for this comment. We hope to have elucidated and answered all your questions correctly.
- The introduction needs to prepared on the basis on the up-to-date literature items
We acknowledge the Reviewer #1 for this comment. We have reviewed our references and change for some more recent articles, we hope that the suggestion has been pertinently answered.
- Conclusions should represent the answer to the research question/tasks specified in the purpose/objectives of the manuscript. However, it is difficult to find a precisely defined purpose of the work in the manuscript of the publication sent to the TOXINSeditorial office. The purpose/aim of the study should be precisely defined and supplemented in the manuscript;
We acknowledge the Reviewer #1 for this comment and inform that we made the change at the end of our introduction, where the objective of our work was exploring and describe the cellular profile and immunological mediators involved in B. atrox envenomation (L58-L61)
- In the Materials and Methods section the inclusion and study inclusion criteria were not clearly included. The lack of mentioned criteria may call into question the control of the test results obtained.
We acknowledge the Reviewer #1 for this comment and inform that we describe in more detail the criteria used for the inclusion of participants in the study, as well as the criteria used for dividing patients into MILD and SEVERE groups (L326-L341).
- Test results do not include routine markers of inflammation such as ESR or CRP. The results of the study do not include enzymatic markers describing the functional state of the liver. It is absolutely necessary to supplement the reviewed manuscript with the mentioned the result of routine laboratory tests or explain the absence of the test results in question;
We acknowledge the Reviewer #1 for this comment and inform that these exams and markers of inflammation for are not requested in the medical routine during the patients' hospitalization period, only in isolated cases and according to the clinical evolution of the patients, for this reason these data are not in the study. We include this information in our work. (L366-L368).
- The manuscript in question omits the practical/clinical implications of the described in this publication results.
We acknowledge the Reviewer #1 for this comment the information presented in this study elucidates this mechanism of the inflammatory response after Bothrops snakebites. Regarding clinical and practical applications, we suggest that further studies be carried out to identify molecules or cells that could be a predictive biomarker in these patients. We appreciate the suggestion and have included the information in our conclusion (L316-L319).
Collectively, we welcome the Reviewers' comments and contributions, and we advise that we have added new sentences and discussions to the work as suggested and requested. Our results explore and demonstrate cell populations and soluble mediators as important components of inflammatory response, with a distinct profile witnessed between groups after the snakebites. We hope that our study can contribute to the elucidation of the immune response after the snakebites, in addition to encouraging further studies in the area.
We believe the new changes have significantly improved the quality of our manuscript. We would like to thank Toxins - Venoms and Toxin-Mediated Local Manifestations members and reviewers for their dedication to providing valuable, and thorough comments on this article. We sincerely hope that the revised version of our manuscript meets the Toxins - Venoms and Toxins - Mediated Local Manifestations high publishing standards and is therefore acceptable for publication in this journal.
With best wishes,
Reviewer 2 Report
Describe the average dosis (number of vials) of antivenom provided in the MILD and SEV groups.
Table 1.
- Describe the type of adverse reactions. Where they severe ones?
- The number of adverse events to antivenom seems too big. Is it what it´s expected. Please, elaborate.
- Could the higher number of adverse reactions in SEV patients as compared to the MILD group be related to the higher dosis used in the former group. Please elaborate.
Author Response
Response Letter
February 18th, 2023
To: Academic Editor
Toxins
Section: Animal Venoms
Special Issue: Venoms and Toxin-Mediated Local Manifestations
Dear Editor,
We very much appreciate the kind consideration given to our manuscript entitled “Exploring the profile of cell populations and soluble immunological mediators in Bothrops atrox envenomation. We hope that the replies to the reviewer’s comments will have satisfactorily improved the manuscript.
Below, we present all the queries made by the reviewers. The changes requested are clearly outlined in the revised manuscript and marked in green. We have prepared a list of answers to the reviewers’ comments, which are highlighted in “bold italic”. In the response to each query, we also included the location of the modified part in the revised manuscript.
Reviewers' comments:
Reviewer reports:
Reviewer #2:
- Describe the average doses (number of vials) of antivenom provided in the MILD and SEV groups.
We thank you and accept the suggestion, the number of vials of Bothrops antivenom was included in the Materials and Methods (L326-L341).
- Describe the type of adverse reactions. Where they severe ones?
We thank you for suggestion Adverse reactions to antivenom, found in MILD and SEVERE groups, were classified as anaphylactic clinical reactions, reactions described in the literature as mild or rare events, such as pruritus and skin reactions, hyperemia, and nausea, and we described them in the caption of the Table 01 (L78-L82).
- The number of adverse events to antivenom seems too big. Is it what it´s expected. Please, elaborate.
We thank for the revealing comment from reviewer #2. The number of adverse reactions found was high, and after reviewing the medical records of each patient, we found that this information is correct. Patients in the SEVERE group had a higher number of adverse events, and we added a paragraph in the discussion, better explaining the reaction (L217-L219).
- Could the higher number of adverse reactions in SEV patients as compared to the MILD group be related to the higher doses used in the former group. Please elaborate.
Thanks to reviewer #2 for the comment. There are studies that demonstrate a greater adverse reaction after severe accidents. After the classification severe of the accident, the patient is treated with 12 vials of antivenom, and this may explain the greater number of these reactions in this group. We added a paragraph about this in the discussion of our work (L217-L219).
Collectively, we welcome the Reviewers' comments and contributions, and we advise that we have added new sentences and discussions to the work as suggested and requested. Our results explore and demonstrate cell populations and soluble mediators as important components of inflammatory response, with a distinct profile witnessed between groups after the snakebites. We hope that our study can contribute to the elucidation of the immune response after the snakebites, in addition to encouraging further studies in the area.
We believe the new changes have significantly improved the quality of our manuscript. We would like to thank Toxins - Venoms and Toxin-Mediated Local Manifestations members and reviewers for their dedication to providing valuable, and thorough comments on this article. We sincerely hope that the revised version of our manuscript meets the Toxins - Venoms and Toxins - Mediated Local Manifestations high publishing standards and is therefore acceptable for publication in this journal.
With best wishes,
Reviewer 3 Report
The manuscript entitled “Exploring the profile of cell populations and soluble immunological mediators in Bothrops atrox envenomations” analyzes the immune response generated in patients envenomated by the viperid snake Bothrops atrox. In the described work, the authors studied envenomated patients before and after treatment with antivenom, also describing the difference between patients with mild and severe manifestations.
I believe the findings are relevant and the work is adequately performed and therefore have only some minor comments that I hope will be useful to the authors:
Line 69. Please clarify the type of adverse reactions that were present and why are these claimed to be caused by antivenom.
Line 79. Please clarify if the # symbol represents a significant difference between the mild and severe groups, it is not completely clear from the table caption.
Figure 3. The similarity of the colors to the previous figures can cause confusion, I suggest using a different color scheme for this figure.
Line 161. There is a mention to red lines that are not present in the figure. Also, same line in the caption speaks about blister exudates that are not present in the figure. I believe this line is mistaken.
Line 323. Given that the differentiation between the Mild and Severe groups is central to the work, I believe it should be described more in depth. Please include a more detailed description of the inclusion criteria for either of these groups.
Line 351. Was there any addition of anticoagulants? If so, please describe.
Line 363. Would it be relevant to include incubation time and temperature?
Line 375. Please describe what was used as erythrocyte lysing solution.
Line 215. Even though there is a mention of the possible implications of age difference between the healthy donors and envenomated patients, there is also an age difference between the Mild and Severe groups. Is this difference significant, and if so, could it have an implication in any of the findings of the work?
Author Response
Response Letter
February 18th, 2023
To: Academic Editor
Toxins
Section: Animal Venoms
Special Issue: Venoms and Toxin-Mediated Local Manifestations
Dear Editor,
We very much appreciate the kind consideration given to our manuscript entitled “Exploring the profile of cell populations and soluble immunological mediators in Bothrops atrox envenomation. We hope that the replies to the reviewer’s comments will have satisfactorily improved the manuscript.
Below, we present all the queries made by the reviewers. The changes requested are clearly outlined in the revised manuscript and marked in green. We have prepared a list of answers to the reviewers’ comments, which are highlighted in “bold italic”. In the response to each query, we also included the location of the modified part in the revised manuscript.
Reviewers' comments:
Reviewer reports:
Reviewer #3:
The manuscript entitled “Exploring the profile of cell populations and soluble immunological mediators in Bothrops atrox envenomations” analyzes the immune response generated in patients envenomated by the viperid snake Bothrops atrox. In the described work, the authors studied envenomated patients before and after treatment with antivenom, also describing the difference between patients with mild and severe manifestations. I believe the findings are relevant and the work is adequately performed and therefore have only some minor comments that I hope will be useful to the authors:
We acknowledge the Reviewer #3 for this comment. We hope to have elucidated and answered all your questions correctly.
- Line 69.Please clarify the type of adverse reactions that were present and why are these claimed to be caused by antivenom.
We acknowledge the Reviewer #3 for this comment. The reactions presented were classified by the medical team as mild or rare, and we described them in the caption of the Table 01 (L78-L82). With the classification severe of the accident, the patient is treated with 12 vails of antivenom, and this may explain the greater number of these reactions in this group. We added a paragraph about this in the discussion of our work (L217 – L219).
- Line 79.Please clarify if the # symbol represents a significant difference between the mild and severe groups, it is not completely clear from the table caption.
We acknowledge the Reviewer #3 for this comment. On Table 01, variables such as gender and age, are compared in the three study groups. ( HD, MILD and SEVERE) However, the significant comparison regarding age was only comparing between MILD and SEVERE groups. The caption of "#" symbol identifies that the significant statistic is only between these 2 groups.
“Significant differences at p<0.05 for comparisons between MILD and SEVERE groups are represented in bold with the superscript symbol #” (L78-L82).
The other variables in the table only exist in the snakebite patients (MILD and SEVERE), for this reason, and for clarity, we include "#" ins significant differences comparisons between these groups. We hope this change makes the table easier to read. We appreciate your contribution.
- Figure 3.The similarity of the colors to the previous figures can cause confusion, I suggest using a different color scheme for this figure.
Thank you for the suggestion, and we have changed the colors of figures 3 and 5. We hope we have reached the suggestion correctly.
- Line 161.There is a mention to red lines that are not present in the figure. Also, same line in the caption speaks about blister exudates that are not present in the figure. I believe this line is mistaken.
We appreciate reviewer #3 suggestion. We inform you that we have made the suggested corrections.
- Line 323.Given that the differentiation between the Mild and Severe groups is central to the work, I believe it should be described more in depth. Please include a more detailed description of the inclusion criteria for either of these groups.
We appreciate reviewer #3 suggestion. In our Materials and Methods, we describe in more detail of inclusion and non-inclusion criteria, in addition we insert the number of antivenom vails received in each group. We hope this makes reading more complete (L326 – L341)
- Line 351.Was there any addition of anticoagulants? If so, please describe.
We thank the reviewer #3 for the suggestion and comment. There was no addition of anticoagulants. We included this information (L369).
- Line 363.Would it be relevant to include incubation time and temperature?
We appreciate the suggestion of Reviewer #3 and consider it pertinent. We inform that after collection, the samples were stored at room temperature and processed within 4 hours (L365-L366). In the immunophenotypic characterization, the samples were incubated at room temperature for 30 minutes. We make changes in the text (L381 – L382).
- Line 375.Please describe what was used as erythrocyte lysing solution.
Thank you, Reviewer #3 for the suggestion. The lysing solution used in our tests is BD FACS™ Lysing Solution (BD® Biosciences San Diego, CA, USA). We performed the description of the preparation of the solution in the Materials and Methods (L393 – L395), we appreciate the comment that enriched our work.
- . Line 215.Even though there is a mention of the possible implications of age difference between the healthy donors and envenomated patients, there is also an age difference between the Mild and Severe groups. Is this difference significant, and if so, could it have an implication in any of the findings of the work?
It could not have an implication on the conclusions of the work. This difference is significant, as shown in Table 01. The patient's age maybe influences their clinical evolution; however, it is worth noting that we used as exclusion criteria patients with comorbidities or immune diseases, which influenced the levels of immunological elements (L326 – L341) during its clinical evolution.
Regarding the possible implications of age difference between the healthy donors and snakebites patients, it is described that older patients tend to be the most affected by snakebites. In addition, accidents mainly involve men, of working age and generally older than the control group.
- Ibiapina HN dos S, Costa AG da, Sachett J de AG, Silva IM, Tarragô AM, Neves JCF, et al. An Immunological Stairway to Severe Tissue Complication Assembly in Bothrops atrox Snakebites. Front Immunol. 2019;10(August):1–12.
- Feitosa ES, Sampaio V, Sachett J, De Castro DB, Noronha M das DN, Lozano JLL, et al. Snakebites as a largely neglected problem in the brazilian amazon: Highlights of the epidemiological trends in the state of amazonas. Rev Soc Bras Med Trop. 2015;48(Suppl I):34–41.
- Wellmann IAM, Ibiapina HNS, Sachett JAG, Sartim MA, Silva IM, Oliveira SS, et al. Correlating Fibrinogen Consumption and Profiles of Inflammatory Molecules in Human Envenomation’s by Bothrops atrox in the Brazilian Amazon. Front Immunol. 2020;11.
Collectively, we welcome the Reviewers' comments and contributions, and we advise that we have added new sentences and discussions to the work as suggested and requested. Our results explore and demonstrate cell populations and soluble mediators as important components of inflammatory response, with a distinct profile witnessed between groups after the snakebites. We hope that our study can contribute to the elucidation of the immune response after the snakebites, in addition to encouraging further studies in the area.
We believe the new changes have significantly improved the quality of our manuscript. We would like to thank Toxins - Venoms and Toxin-Mediated Local Manifestations members and reviewers for their dedication to providing valuable, and thorough comments on this article. We sincerely hope that the revised version of our manuscript meets the Toxins - Venoms and Toxins - Mediated Local Manifestations high publishing standards and is therefore acceptable for publication in this journal.
With best wishes,
Round 2
Reviewer 1 Report
Authors - referred to the reviewer's comments and made corrections in the manuscript of the publication.
In my - current assessment - the article is ready for publication.